# DreamID-Omni: Unified Framework for Controllable Human-Centric Audio-Video Generation

Xu Guo [* 1]   Fulong Ye [* 2]   Qichao Sun [* 2]   Liyang Chen [1]   Bingchuan Li [2]   Pengze Zhang [2]   Jiawei Liu [2]
Songtao Zhao [2]   Qian He [2]   Xiangwang Hou [1]

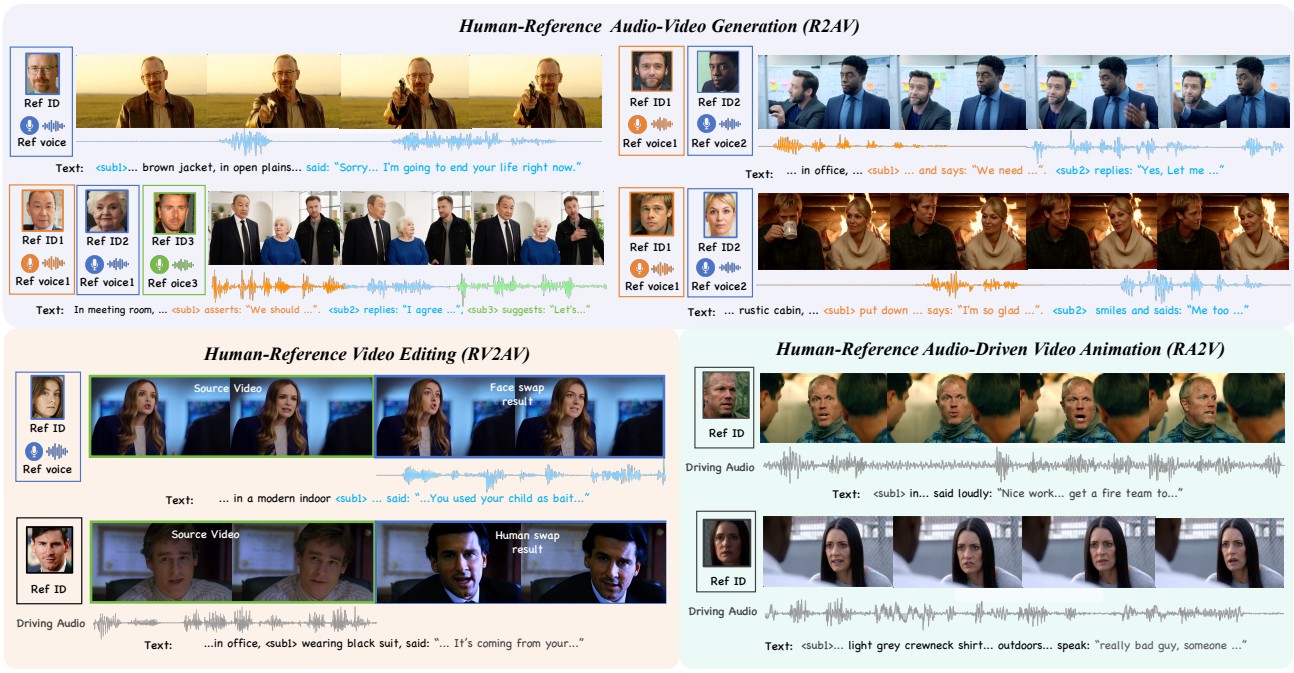

*Figure 1.* **Showcase of *DreamID-Omni*.** *DreamID-Omni* seamlessly unifies human reference-based audio-video generation (R2AV), video editing (RV2AV), and audio-driven video animation (RA2V).

## Abstract

Recent advancements in foundation models have revolutionized joint audio-video generation. However, existing approaches typically treat human-centric tasks including reference-based audio-video generation (R2AV), video editing (RV2AV) and audio-driven video animation (RA2V) as isolated objectives. Furthermore, achieving precise, disentangled control over multiple character identities and voice timbres within a single framework remains an open challenge. In this paper, we propose *DreamID-Omni*, a unified framework for controllable human-centric audio-video gen-

eration. Specifically, we design a Symmetric Conditional Diffusion Transformer that integrates heterogeneous conditioning signals via a symmetric conditional injection scheme. To resolve the pervasive identity-timbre binding failures and speaker confusion in multi-person scenarios, we introduce a Dual-Level Disentanglement strategy: Synchronized RoPE at the signal level to ensure rigid attention-space binding, and Structured Captions at the semantic level to establish explicit attribute-subject mappings. Furthermore, we devise a Multi-Task Progressive Training scheme that leverages weakly-constrained generative priors to regularize strongly-constrained tasks, preventing overfitting and harmonizing disparate objectives. Extensive experiments demonstrate that *DreamID-Omni* achieves comprehensive state-of-the-art performance across video, audio, and audio-visual consistency, even outperforming leading proprietary commercial models.

[1]Tsinghua University [2]ByteDance. Correspondence to: Xiangwang Hou <xwhou@mail.tsinghua.edu.cn>, Songtao Zhao <zhaosongtao.0815@bytedance.com>.

*Proceedings of the 43rd International Conference on Machine Learning*, Seoul, South Korea. PMLR 306, 2026. Copyright 2026 by the author(s).

# 1. Introduction

Recently, joint audio-video generation has seen rapid progress, with many breakthrough works emerging. For example, commercial models such as Veo3, Sora2, Wan 2.6 (Wan et al., 2025) and Seedance 1.5 Pro (Chen et al., 2025c) have achieved impressive results. In the open-source community, models like Ovi (Low et al., 2025) and LTX-2 (HaCohen et al., 2026) have also demonstrated promising performance. These advances have greatly promoted the development of joint audio-video generation. However, in real-world applications, supporting more controllable generation particularly within human-centric scenarios is crucial.

Controllable human-centric generation has advanced in several directions. Works such as Phantom (Liu et al., 2025b) and Wan2.6 (Wan et al., 2025) utilize reference images or voice timbres for video (R2V) or audio-video (R2AV) generation, which rely solely on text prompts as a weakly-constrained guidance. To achieve higher controllability, other approaches introduce stronger supervision, such as source videos or driving audio, for strongly-constrained generation. For instance, Humo (Chen et al., 2025b) animates videos (RA2V) based on reference identities and driving audio, while works like HunyuanCustom (Hu et al., 2025) and VACE (Jiang et al., 2025) perform video editing given a reference identity and source video, which can be further extended to replace the corresponding audio (RV2AV). Despite these advancements, these capabilities are largely treated as isolated tasks. Researchers in the video-only domain have begun to shift toward unified architectures (Jiang et al., 2025; Ye et al., 2025; Liang et al., 2025; Yang et al., 2025; Qu et al., 2025; He et al., 2025) to enhance task flexibility and reduce the operational overhead of deploying multiple models. However, the joint audio-video domain still lacks a unified perspective. Fundamentally, we observe that R2AV, RV2AV, and RA2V all share an identical objective: mapping a static identity anchor (image and audio) onto a dynamic spatio-temporal canvas (text, source video, or driving audio). Based on this insight, these tasks are inherently amenable to a unified framework trained on a consistent data source, transcending the limitations of task-specific silos.

Nevertheless, developing this unified framework presents several challenges: (1) How to build a unified model framework that supports generation, editing and animation; (2) How to address identity-timbre binding and speaker confusion in multi-person generation; (3) How to design effective training strategies to prevent conflicts among multiple tasks.

To address these challenges, we introduce *DreamID-Omni*, which integrates reference-based generation, editing, and animation into a single paradigm. *DreamID-Omni* builds upon a dual-stream Diffusion Transformer (Peebles & Xie, 2022) (DiT) architecture, where video and audio streams interact via bidirectional cross-attention for fine-grained synchronization. We propose a Symmetric Conditional DiT design that unifies heterogeneous conditioning signals—reference images, voice timbres, source videos, and driving audio—into a shared latent space, enabling seamless task switching without architectural changes.

To resolve multi-person confusion, we propose a Dual-Level Disentanglement strategy. At the signal level, Synchronized Rotary Positional Embeddings (Syn-RoPE) is introduced to bind reference identities with their corresponding voice timbres within the attention space. At the semantic level, Structured Captions utilize anchor tokens paired with fine-grained descriptions to establish explicit mappings between specific subjects and their respective attributes or speech content.

Finally, we devise a Multi-Task Progressive Training strategy to harmonize the three tasks. In the initial two stages, we focus exclusively on the weakly-constrained R2AV task, employing in-pair reconstruction and cross-pair disentanglement to enhance identity and timbre fidelity while encouraging the model to learn robust reference representations. In the final stage, strongly-constrained tasks (RV2AV and RA2V) are introduced for joint training with R2AV. This approach prevents the model from overfitting to strongly-constrained tasks, thereby maintaining superior performance on the weakly-constrained generation task.

In summary, our contributions are as follows: (1) We propose *DreamID-Omni*, a novel human-centric controllable generation framework based on a Symmetric Conditional DiT, which seamlessly integrates R2AV, RV2AV, and RA2V tasks. (2) We introduce Dual-Level Disentanglement, which addresses identity-timbre binding and speaker confusion in multi-person generation via Syn-RoPE and Structured Captions. (3) We present a Multi-Task Progressive Training strategy that effectively harmonizes diverse tasks with varying constraint strengths. (4) Extensive experiments demonstrate that *DreamID-Omni* achieves comprehensive state-of-the-art performance across video, audio, and audio-visual consistency, even when compared to leading proprietary commercial models.

# 2. Related Work

## 2.1. Joint Audio-Video Generation

Recent advancements in diffusion-based foundation models in video generation (Wan et al., 2025; Kong et al., 2024; Gao et al., 2025) and audio generation (Liu et al., 2023; Gong et al., 2025) have significantly expanded the frontier of joint audio-video synthesis. While pioneering works (Ruan et al., 2023) use coupled U-Net backbones, current DiT-based approaches dominate the field. These methods typically employ either dual-stream architectures (Liu et al., 2024; Hayakawa et al., 2024; Wang et al., 2025a; Liu et al., 2025a;

Low et al., 2025; HaCohen et al., 2026) with specialized fusion layers (e.g., cross-attention) or unified DiT structures (Wang et al., 2024; Zhao et al., 2025a; Huang et al., 2025a; Wang et al., 2026) with joint self-attention to achieve synchronized multi-modal alignment. Despite their impressive generative fidelity, these models are primarily designed for vanilla text-to-audio-video or first-frame-conditioned synthesis. They lack the capability to condition the generative process on external identity or voice timbre references. This limitation restricts their utility in scenarios requiring persistent identity and timbre consistency.

### 2.2. Controllable Video Generation Model

**Reference-based Generation.** To enhance controllability, reference-based video generation has emerged as a prominent research direction, focusing on maintaining identity consistency by integrating reference features into the diffusion process. While initial efforts (He et al., 2024; Yuan et al., 2025; Polyak et al., 2024) were primarily tailored for single-identity scenarios, subsequent research has extended these capabilities to multi-subject settings (Zhong et al., 2025; Huang et al., 2025b; Chen et al., 2025a; Liu et al., 2025b; Hu et al., 2025; Li et al., 2025; Deng et al., 2025). However, these works are typically video-centric and do not support audio generation.

**Video Editing and Animation.** In terms of temporal control, tasks can be categorized into video editing and audio-driven video animation. Editing frameworks (Chen et al., 2024; Guo et al., 2026; Luo et al., 2025; Wang et al., 2025c; Shao et al., 2025; Jiang et al., 2025; Xu et al., 2026; Cheng et al., 2025) allow for the modification of identity attributes within the source video. Audio-driven video animation (Wei et al., 2024; Xu et al., 2024; Chen et al., 2025b; Wang et al., 2025b; Lin et al., 2025) aims to generate videos from reference images to produce lip movements matching input speech signals. Despite their success, these models are all task-specific, and no existing model attempts to unify reference-based generation, editing, and animation.

## 3. Methodology

### 3.1. Problem Formulation

We unify the landscape of controllable human-centric generation into a single probabilistic framework. Given a text prompt $\mathcal{T}$, a set of reference identities $\mathcal{I} = \{I_1, \ldots, I_N\}$, and corresponding reference voice timbres $\mathcal{A} = \{A_1, \ldots, A_N\}$, the goal is to synthesize a synchronized video-audio stream $Y = \{Y_{\text{video}}, Y_{\text{audio}}\}$.

To support reference-based editing and animation tasks, we introduce two optional structural conditions: a source video context $V_{\text{src}}$ and a driving audio stream $A_{\text{dri}}$. The framework

models the conditional distribution:

$$P(Y \mid \mathcal{T}, \mathcal{I}, \mathcal{A}, V_{\text{src}}, A_{\text{dri}}) \qquad (1)$$

By selectively providing these conditions, our framework seamlessly transitions between three distinct tasks, as summarized in Table 1.

*Table 1.* **Task Unification in *DreamID-Omni*.** Our framework unifies Human-Reference Audio-Video Generation (R2AV), Human-Reference Video Editing (RV2AV) and Human-Reference Audio-Driven Video Animation (RA2V) by toggling input conditions.

| Task | Input | Output Goal |
|------|-------|-------------|
| R2AV | $\mathcal{T}, \mathcal{I}, \mathcal{A}$ | Generate with references $\mathcal{I}, \mathcal{A}$. |
| RV2AV | $\mathcal{T}, \mathcal{I}, \mathcal{A}, V_{\text{src}}$ | Edit identity and audio in $V_{\text{src}}$. |
| RA2V | $\mathcal{T}, \mathcal{I}, A_{\text{dri}}$ | Animate identity $\mathcal{I}$ using $A_{\text{dri}}$. |

### 3.2. Framework

To address the diverse tasks defined in Section 3.1, we propose *DreamID-Omni*, a unified framework built upon a dual-stream DiT, as illustrated in Figure 2. The architecture consists of two parallel backbones: a video stream for visual synthesis and an audio stream for acoustic synthesis. These streams interact via bidirectional cross-attention layers, enabling fine-grained temporal synchronization and semantic alignment between the visual and auditory modalities.

#### 3.2.1. SYMMETRIC CONDITIONAL DiT

A core architectural contribution of *DreamID-Omni* is the Symmetric Conditional DiT, designed to seamlessly integrate reference-based generation, editing, and animation within a unified framework. This is achieved through a symmetric dual-stream conditioning strategy that composes heterogeneous control signals in the latent space with structural parity. Let $z_v$ and $z_a$ represent the noisy target video and target audio latents, respectively. To guide the denoising process, we construct two comprehensive conditional sequences, $X_v$ and $X_a$, which integrate both identity-specific and structural guidance:

$$X_v = [z_v; \mathcal{E}_v(\mathcal{I})] + [\mathcal{E}_v(V_{\text{src}}); \mathbf{0}_{\mathcal{E}_v(\mathcal{I})}] \qquad (2)$$

$$X_a = [z_a; \mathcal{E}_a(\mathcal{A})] + [\mathcal{E}_a(A_{\text{dri}}); \mathbf{0}_{\mathcal{E}_a(\mathcal{A})}] \qquad (3)$$

where $[\cdot; \cdot]$ denotes concatenation along the sequence dimension, $\mathbf{0}_T$ represents a zero tensor with the same shape as $T$, and $\mathcal{E}_v, \mathcal{E}_a$ are the respective VAE encoders. In this symmetric formulation, the reference features ($\mathcal{E}_v(\mathcal{I}), \mathcal{E}_a(\mathcal{A})$) are concatenated to the noisy latents, allowing the DiT blocks to extract and disentangle high-level identity and timbre priors. Simultaneously, the structural conditions ($V_{\text{src}}, A_{\text{dri}}$) are injected via element-wise addition, serving as a structural canvas that enforces spatial and temporal consistency.

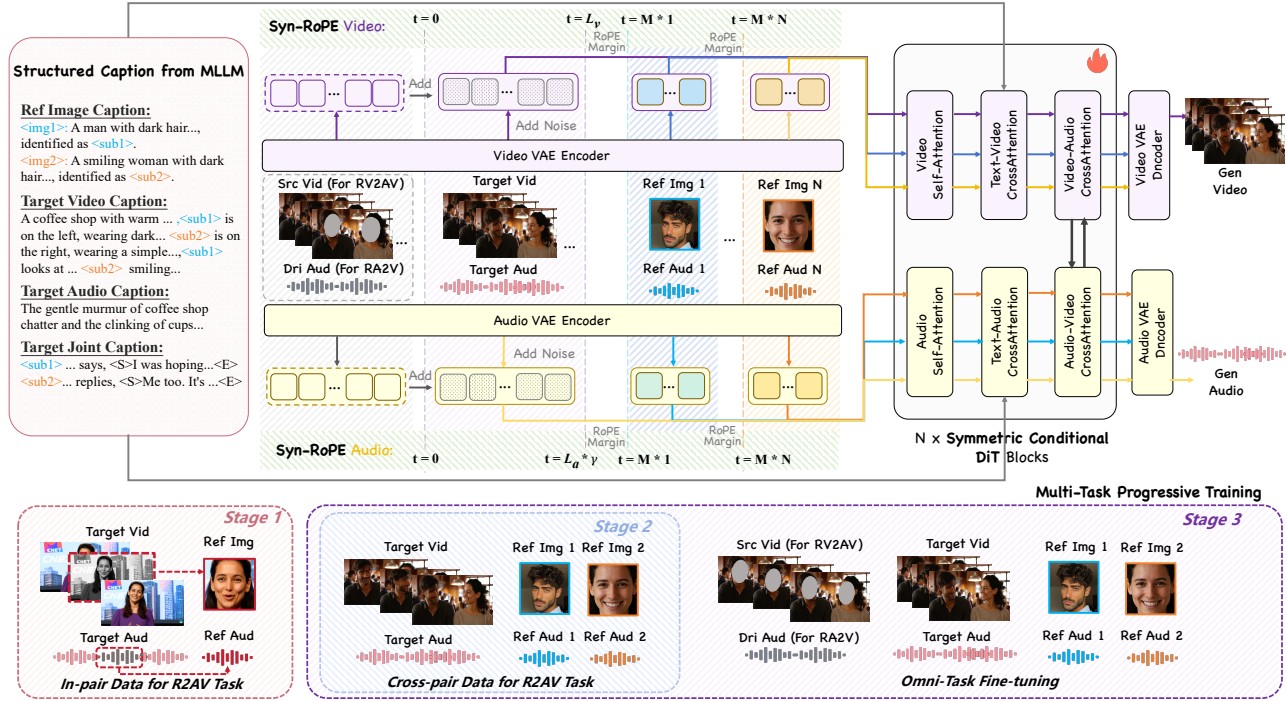

*Figure 2.* **Overview of *DreamID-Omni* framework.** We integrate reference-based generation (R2AV), editing (RV2AV), and animation (RA2V) using a Symmetric Conditional DiT trained via a multi-task progressive training strategy. Structured Caption and Syn-RoPE ensure robust dual-level disentanglement in multi-person scenarios.

This dual-injection strategy effectively decouples the conditioning into identity-preservation and structural-guidance channels.

The inherent flexibility of this design enables seamless task switching. As detailed in Table 1, providing a null input for the structural conditions ($V_{\text{src}}$ or $A_{\text{dri}}$) effectively nullifies the additive term in Eqs. 2 or 3. Consequently, the model adaptively transitions between R2AV, RV2AV, and RA2V based on the available conditional modalities, maintaining a unified parameter set across all functional modes.

### 3.2.2. DUAL-LEVEL DISENTANGLEMENT

A critical challenge in multi-person generation is the confusion between subjects, which manifests in two forms: identity-timbre mismatch (e.g., subject A speaks with the voice of subject B) and attribute-content misattribution (e.g., subject A erroneously inheriting the visual attributes and dialogue of subject B). We posit that these failures stem from entanglement at two distinct levels. At the **signal level**, standard attention mechanisms fail to bind the visual features of an identity to its corresponding voice timbre. At the **semantic level**, unstructured text captions provide insufficient granularity to explicitly link specific subjects to their respective visual attributes, motions, and speech content. To address this, we propose a Dual-Level Disentanglement strategy. We introduce Syn-RoPE to enforce a rigid binding at the signal level, and a Structured Captioning scheme to

resolve ambiguity at the semantic level.

**Syn-RoPE.** Recent works (Kong et al., 2025) have explored using Rotary Position Embedding (Su et al., 2024) for spatial localization within video frames. However, such a spatially-grounded approach is incompatible with the more challenging task like R2AV, where character positions are synthesized dynamically by the model. To overcome this limitation, we propose Syn-RoPE, an identity-grounded mechanism that operates by assigning distinct, non-overlapping temporal positional segments to different semantic inputs within the model's attention space. As illustrated in Figure 2, inspired by (Low et al., 2025), we synchronize the video and audio streams by scaling the RoPE frequencies of the target audio latents by a factor $\gamma = L_v/L_a$, where $L_v$ and $L_a$ denote the sequence lengths of the target video and audio latents, respectively. More crucially, Syn-RoPE partitions the absolute temporal positional index space into reserved "RoPE Margins" for the target sequence and each reference identity. Specifically, the target video and audio latents occupy the initial positional range $[0, L-1]$, where $L$ denotes the maximum temporal length. We define a fixed margin $M$ such that $M \gg L$ to serve as the base interval for each identity slot. Subsequently, the latent features of the $k$-th reference identity (both image $\mathcal{I}_k$ and audio $\mathcal{A}_k$) are assigned to the $k$-th reserved segment, $[k \cdot M, (k+1) \cdot M - 1]$. This strategy offers two fundamental advantages: (i) **Inter-Identity Decoupling:** By leveraging the periodicity of RoPE, each identity's features are projected into a distinct rotational sub-

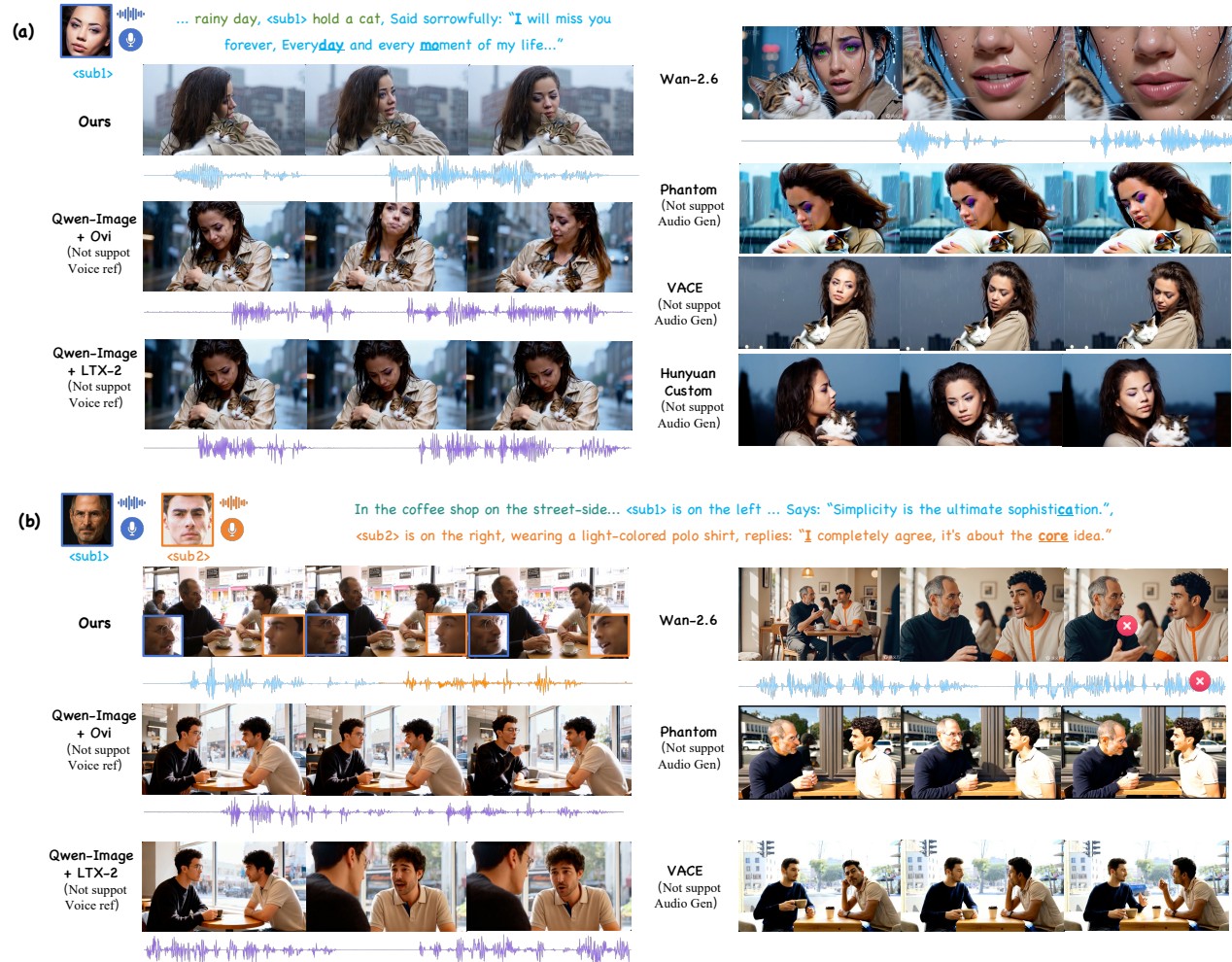

*Figure 3.* **Qualitative comparison** with state-of-the-art methods on R2AV. Please zoom in for more details.

space, naturally suppressing cross-identity attention scores and preventing feature entanglement. (ii) **Intra-Identity Synchronization:** By mapping the visual and acoustic features of the same identity to identical positional segments, we achieve a robust, implicit cross-modal synchronization at the signal level. This design provides a unified mechanism for robust identity binding across all generation, editing, and animation tasks.

**Structured Caption.** At the semantic level, ambiguity in multi-subject scenarios typically arises when standard prompts fail to explicitly associate visual attributes, motions, and speech content with specific individuals. To resolve this, we introduce a Structured Captioning scheme that establishes an unambiguous mapping between each reference identity $\mathcal{I}_k$ and a unique **anchor token**, denoted as $\langle sub_k \rangle$. The process begins by generating a fine-grained attribute description for each identity to initialize the anchor tokens. Building upon this foundation, the target video content is synthesized into a comprehensive "script" partitioned into distinct semantic fields: *video caption*, *audio caption*, and

*joint caption*. Crucially, all references to individuals across these fields consistently utilize the predefined anchor tokens $\langle sub_k \rangle$. This format provides the model with an explicit grounding that resolves semantic-level entanglement, which is critical for the success of all three core tasks.

### 3.3. Multi-Task Progressive Training

Training a unified model for R2AV, RV2AV, and RA2V presents a complex optimization challenge. A naive joint training approach often suffers from conflicting learning objectives, where the generative objective of creating diverse content can interfere with the fidelity objective of adhering to strong conditional constraints. To circumvent this, we introduce a Multi-Task Progressive Training Strategy, a three-stage curriculum designed to incrementally build model capabilities, ensuring stable convergence and synergistic learning.

**In-pair Reconstruction.** The initial stage aims to establish a robust generative prior for controllable generation. We

train the model exclusively on the R2AV task, using an in-pair reconstruction objective. For each training sample $Y$, we extract the reference identity $\mathcal{I}$ and the reference timbre $\mathcal{A}$ from the sample itself. The model is then tasked with reconstructing the full data stream conditioned on these internal references and the text prompt $\mathcal{T}$. To prevent the model from trivially copying the reference segments and to encourage true conditional synthesis, we introduce a masked reconstruction loss. Let $\mathcal{M}_v$ and $\mathcal{M}_a$ be binary masks identifying the spatio-temporal regions of $\mathcal{I}$ and $\mathcal{A}$ within the ground truth latents. The loss is computed only on the unmasked regions, forcing the model to generate, rather than merely copy, the content corresponding to the references. The objective is defined as:

$$
\begin{aligned}
\mathcal{L}_{\text{inpair}} = \mathbb{E}_{z,t,\mathcal{C}} \big[ & \\
\lambda_v \| (1 - \mathcal{M}_v) & \odot (\epsilon_v - \hat{\epsilon}_\theta(z_{v,t}, t, \mathcal{C})) \|_2^2 \quad (4) \\
+ \lambda_a \| (1 - \mathcal{M}_a) & \odot (\epsilon_a - \hat{\epsilon}_\theta(z_{a,t}, t, \mathcal{C})) \|_2^2 \big]
\end{aligned}
$$

where the conditioning set for this stage is $\mathcal{C} = \{\mathcal{I}, \mathcal{A}, \mathcal{T}\}$, $\epsilon$ is the ground truth noise, $\hat{\epsilon}_\theta$ is the model's prediction, and $\odot$ denotes element-wise multiplication.

**Cross-pair Disentanglement.** To enhance the model's generalization capabilities and force it to learn a truly disentangled representation of identity and timbre, we advance to a cross-pair training stage. In this phase, the reference identity $\mathcal{I}$ and timbre $\mathcal{A}$ are sourced from a different video clip than the target video-audio stream $Y$. This more challenging objective compels the model to synthesize content based on abstract identity and timbre concepts, rather than relying on low-level correlations present in the source. The training objective for this stage, $\mathcal{L}_{\text{cross}}$, reuses the same formulation as $\mathcal{L}_{\text{inpair}}$ (Eq. 4). However, a key distinction is that the masks are nullified by setting $\mathcal{M}_v = \mathbf{0}$ and $\mathcal{M}_a = \mathbf{0}$. This modification ensures the loss is computed over the entire data stream, pushing the model towards a more robust disentanglement.

**Omni-Task Fine-tuning.** The final stage unifies all tasks by fine-tuning the model on a mixed dataset comprising R2AV, RV2AV, and RA2V samples. RV2AV samples are constructed by providing a masked version of the target video as structural context ($V_{\text{src}}$), while RA2V samples supply the target audio as the driving signal ($A_{\text{dri}}$). By training on this composite dataset, the model learns to seamlessly switch between generation, editing, and animation based on the provided conditions, as formulated in Eq. 1. This progressive, three-stage curriculum is crucial. We observe that by first mastering the weakly-constrained R2AV task, the model develops a powerful and diverse generative prior. This prior then serves as a robust foundation for the strongly-constrained RV2AV and RA2V tasks, allowing the model to learn high-fidelity conditional control without sacrificing generative quality, leading to a truly unified and capable omni-purpose model.

### 3.4. Inference Pipeline

At inference time, we employ a multi-condition Classifier-Free Guidance (CFG) (Ho & Salimans, 2022) strategy, which is applied independently to the video and audio streams, but follows the same unified formulation:

$$
\begin{aligned}
\hat{\epsilon}_{\text{final}} = \hat{\epsilon}_\theta(z_t, \emptyset, \emptyset) & \\
+ w_\mathcal{T} \cdot (\hat{\epsilon}_\theta(z_t, \mathcal{T}, \emptyset) & - \hat{\epsilon}_\theta(z_t, \emptyset, \emptyset)) \quad (5) \\
+ w_\mathcal{S} \cdot (\hat{\epsilon}_\theta(z_t, \mathcal{T}, \mathcal{S}) & - \hat{\epsilon}_\theta(z_t, \mathcal{T}, \emptyset))
\end{aligned}
$$

where $\hat{\epsilon}_\theta(z_t, \mathcal{T}, \mathcal{S})$ is the model's prediction under text condition $\mathcal{T}$ and a stream-specific condition $\mathcal{S}$. For the video stream, $\mathcal{S} = \mathcal{I}$, while for the audio stream, $\mathcal{S} = \mathcal{A}$. The terms $w_\mathcal{T}$ and $w_\mathcal{S}$ are their respective guidance scales. This chained application ensures that identity and timbre guidance operates on a text-aligned basis, leading to more stable and coherent results. The MLLM system prompt for Structured Caption is shown in Fig. 9.

## 4. Experiments

### 4.1. Setup

**IDBench-Omni.** We introduce *IDBench-Omni*, a new comprehensive benchmark for controllable human-centric audio-video generation. The benchmark comprises three specialized test sets, totaling 200 high-quality data instances, designed to evaluate a model's omni-purpose capabilities: (1) 100 identity-timbre-caption triplets for evaluating generation task; (2) 50 masked videos with target identity and timbre for evaluating controlled video editing; and (3) 50 driving audios with reference identities for evaluating audio-driven animation. These sets cover a diverse range of challenging scenarios, including complex multi-person dialogues, significant variations in identity and timbre, and in-the-wild recording conditions. IDBench-Omni provides a rigorous and holistic platform for evaluating the generation, editing, and animation capabilities of unified audio-video models.

**Implementation Details.** We initialize our model from Ovi (Low et al., 2025) and fully train on audio-video data from (Li et al., 2024) (construction details in Sec. A.2). During training, we set the learning rate to $1.0 \times 10^{-5}$, with a global batch size of 32 and Rope Margin $M = 150$. The training curriculum begins with the In-pair Reconstruction for 10,000 steps, followed by the Cross-pair Disentanglement and Omni-Task Fine-tuning stages, which involve 20,000 iterations each. In the final Omni-Task stage, we sample R2AV, RV2AV, and RA2V data with a ratio of 4:3:3. The total training cost is approximately 128 H20 GPU-days.

**Evaluation Metrics.** We evaluate our model across three key dimensions. **For video**, we assess fidelity and coherence

*Table 2.* Quantitative comparison of R2AV on our proposed benchmark. Best results are in **bold**, second best are underlined. The S/M notation in ID-Sim. and T-Sim. refers to results on single-person and multi-person scenarios, respectively.

| Method | Support | | Video | | | Audio | | | | Audio-Visual Consistency | | |
|---|---|---|---|---|---|---|---|---|---|---|---|---|
| | Video | Audio | AES ↑ | ViCLIP ↑ | ID-Sim. (S/M) ↑ | PQ ↑ | CLAP ↑ | WER ↓ | T-Sim. (S/M) ↑ | Sync-C ↑ | Sync-D ↓ | Spk-Conf. ↓ |
| Phantom | ✓ | ✗ | 0.604 | 13.791 | 0.657/0.572 | - | - | - | - | - | - | - |
| VACE | ✓ | ✗ | 0.613 | 11.091 | 0.664/0.395 | - | - | - | - | - | - | - |
| HunyuanCustom | ✓ | ✗ | 0.589 | 12.159 | 0.659/- | - | - | - | - | - | - | - |
| Qwen-Image + LTX-2 | ✓ | ✓ | 0.611 | 8.548 | 0.571/0.349 | 6.247 | 0.144 | 0.093 | - | 3.706 | 10.003 | 0.340 |
| Qwen-Image + Ovi | ✓ | ✓ | 0.606 | 8.974 | 0.459/0.336 | 5.826 | 0.203 | 0.097 | - | 5.857 | 8.407 | 0.380 |
| Wan2.6 | ✓ | ✓ | **0.632** | 13.410 | 0.523/0.455 | **6.391** | 0.236 | 0.534 | 0.391/0.217 | 6.026 | 8.352 | 0.380 |
| **Ours** | ✓ | ✓ | 0.618 | **13.911** | **0.674/0.603** | 6.290 | **0.278** | **0.052** | **0.493/0.402** | **6.226** | **7.791** | **0.080** |

*Table 3.* Comparison with state-of-the-art methods on RV2AV.

| Method | AES ↑ | ViCLIP ↑ | ID-Sim. ↑ | WER ↓ | T-Sim. ↑ | Sync-C ↑ |
|---|---|---|---|---|---|---|
| VACE | 0.560 | 14.353 | 0.565 | - | - | - |
| HunyuanCustom | 0.538 | 14.576 | 0.590 | - | - | - |
| **Ours** | **0.584** | **14.832** | **0.635** | **0.065** | **0.513** | **6.241** |

*Table 4.* Comparison with state-of-the-art methods on RA2V.

| Method | AES ↑ | ViCLIP ↑ | ID-Sim. ↑ | Sync-C ↑ | Sync-D ↓ |
|---|---|---|---|---|---|
| Humo | 0.550 | 14.859 | 0.609 | 6.114 | **8.323** |
| HunyuanCustom | 0.567 | 13.027 | 0.611 | 5.786 | 9.071 |
| **Ours** | **0.591** | **16.618** | **0.623** | **6.325** | 8.659 |

through the aesthetics score (AES) from VBench (Huang et al., 2024) for video quality, the text-video similarity from ViCLIP (Wang et al., 2023) for text following, and Arc-Face (Deng et al., 2019) for Identity Similarity (ID-Sim.). **For audio**, we evaluate its quality and fidelity from multiple aspects. We gauge audio quality via the Production Quality (PQ) score from AudioBox-Aesthetics (Tjandra et al., 2025) and semantic consistency using CLAP (Wu et al., 2023). Additionally, we compute the Word Error Rate (WER) by transcribing the generated audio with Whisper-large-v3 (Radford et al., 2023) and comparing it against the ground-truth transcript, while Timbre Similarity (T-Sim.) is determined by the cosine similarity of speaker embeddings from WavLM (Chen et al., 2021). **For audio-visual consistency**, we focus on synchronization and attribution. Lip-sync accuracy relies on the standard confidence (Sync-C) and distance (Sync-D) scores from SyncNet (Chung & Zisserman, 2016). Finally, Speaker Confusion (Spk-Conf.), a critical metric for multi-person dialogues, is evaluated by the Gemini-2.5-Pro model (Team et al., 2023), a detailed system prompt provided in Sec. A.3.

## 4.2. Comparison

**Comparison on R2AV.** As there are no open-source methods that directly support the R2AV task, we establish a set of strong baselines for comparison. We compare our method with the closed-source model Wan2.6 (Wan et al., 2025) and two cascaded pipelines constructed by first generating an initial frame with Qwen-Image (Wu et al., 2025) and then animating it with LTX-2 (HaCohen et al., 2026) and Ovi (Low et al., 2025). Additionally, for video-centric metrics, we include leading R2V models: Phantom (Liu

et al., 2025b), VACE (Jiang et al., 2025), and Hunyuan-Custom (Hu et al., 2025). As demonstrated in Table 2, our method achieves superior or comparable results across the video, audio, and audio-visual consistency dimensions. For qualitative comparison in Fig. 3, in case (a), our model delivers the most realistic visual results compared to baselines such as Wan2.6, while exhibiting superior identity consistency with the reference identities relative to Ovi and LTX-2. In case (b), only ours successfully achieves correct binding between specific identities and their corresponding timbres, whereas baselines like Wan2.6 suffer from identity-timbre mismatch. See Sec. A.4 for user study details.

**Comparison on RV2AV.** We compare our method with SOTA video editing methods, VACE (Jiang et al., 2025) and HunyuanCustom (Hu et al., 2025) on RV2AV. The quantitative results are presented in Table 3. Since the compared methods do not support audio generation, audio-related metrics are reported exclusively for our model. The results demonstrate that our method not only achieves SOTA performance on video-centric metrics (AES, ViCLIP, and ID-Sim.), but also exhibits excellent audio generating capabilities, as evidenced by the strong WER, T-Sim., and Sync-C scores. Qualitative results are illustrated in Fig. 5. In case (a), our model delivers higher identity similarity and superior visual quality; in case (b), it demonstrates improved text-following capabilities compared to the baselines.

**Comparison on RA2V.** For the RA2V task, we compare our method with Humo (Chen et al., 2025b) and Hunyuan-Custom (Hu et al., 2025). As shown in Table 4, our method achieves comparable lip-sync accuracy to Humo and leading performance on video-related metrics. Qualitative comparisons are provided in Fig. 6. Notably, in scenarios involving multiple subjects, both Humo and HunyuanCustom frequently exhibit speaker misattribution errors. In contrast, our model animates the correct subject by precisely following the structured captions.

## 4.3. Ablation Studies

**Ablation on Dual-level Disentanglement.** To validate the effectiveness of our dual-level disentanglement design, we conduct an ablation study on the challenging multi-person

*Table 5.* Ablation study on Dual-level Disentanglement.

| Method | ViCLIP ↑ | T-Sim. ↑ | Sync-C ↑ | Sync-D ↓ | Spk-Conf. ↓ |
|---|---|---|---|---|---|
| w/o Syn-RoPE | 13.179 | 0.211 | 4.192 | 10.411 | 0.12 |
| w/o SC | 11.381 | 0.378 | 5.943 | 8.064 | 0.26 |
| **Ours** | **13.613** | **0.402** | **6.074** | **8.027** | **0.08** |

*Table 6.* Ablation study on Multi-Task Progressive Training.

| Method | ViCLIP ↑ | ID-Sim.↑ | AQ ↑ | T-Sim. ↑ | CLAP ↑ |
|---|---|---|---|---|---|
| Only IR | 11.931 | **0.692** | 5.576 | **0.504** | 0.225 |
| Only CD | 14.044 | 0.543 | 6.072 | 0.471 | **0.287** |
| MT (w/o OFT) | 9.518 | 0.638 | 4.449 | 0.442 | 0.104 |
| **Ours** | **14.573** | 0.674 | **6.313** | 0.493 | 0.282 |

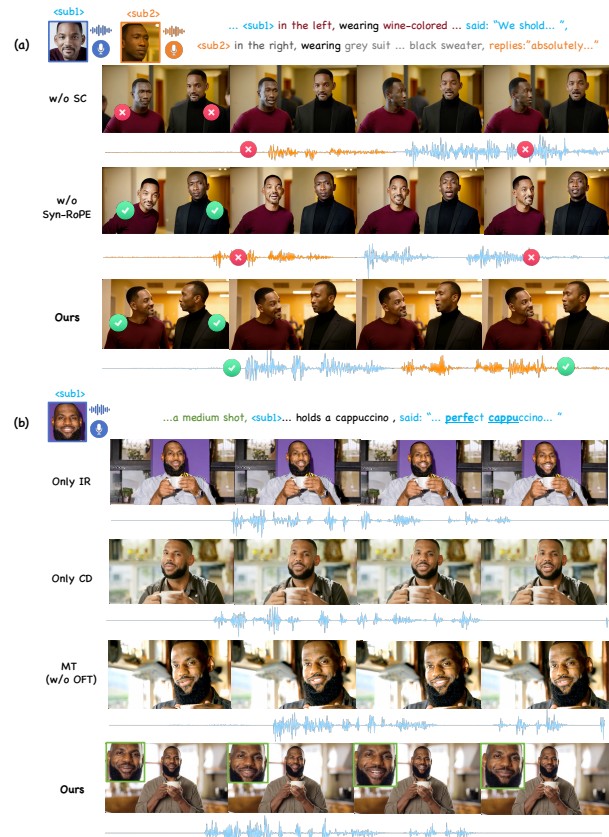

*Figure 4.* **Qualitative results of our ablation studies.** (a) Ablation on our dual-level disentanglement design. (b) Ablation on multi-task progressive training.

dialogue scenario of the R2AV task. The quantitative results are presented in Table 5, with a qualitative comparison in Fig. 4 (a). Our analysis highlights the distinct contributions of each component: (1)**w/o SC:** Following Ovi (Low et al., 2025), we replace the Structured Caption (SC) with standard unstructured joint caption (i.e., a single global description for both target video and audio), the model's ability to follow textual instructions is significantly impaired, resulting in the lowest ViCLIP score. More critically, this leads to a dramatic increase in speaker confusion, with the Spk-Conf. rate more than tripling from 0.08 to 0.26. This underscores the crucial role of SC in explicitly associating visual attributes and dialogue content with specific subjects in multi-person scenarios. As illustrated in the first row of Fig. 4 (a), without SC, both the visual attributes and content of $\langle sub_1 \rangle$ and $\langle sub_2 \rangle$ suffer from severe mismatch. (2)**w/o Syn-RoPE:** Removing Syn-RoPE, which is designed to bind specific speakers to their corresponding timbres, leads to a severe degradation in timbre preservation, as indicated by the sharp drop in the T-Sim. score. The identity-timbre mismatch also negatively impacts lip-sync accuracy (Sync-C/D). As shown in the second row of Fig. 4 (a), without Syn-RoPE, $\langle sub_1 \rangle$ is erroneously bound to the voice timbre of $\langle sub_2 \rangle$.

**Ablation on Multi-Task Progressive Training.** We conduct an ablation study on our multi-task progressive training strategy, with the results on the single-person R2AV scenario presented in Table 6. A qualitative comparison is provided in Fig. 4 (b) **Only IR:** Training exclusively with In-pair Reconstruction (IR) leads to severe copy-paste issues, as illustrated in Fig. 4 (b) (the first row). While this results in deceptively high ID-Sim. and T-Sim. scores, the model fails to learn meaningful conditional synthesis, leading to poor text-following ability (ViCLIP) and audio quality (AQ). (2) **Only CD:** Conversely, training only with Cross-pair Disentanglement (CD) from the start proves too challenging. The model struggles to learn fundamental representations, resulting in very low ID-Sim. and T-Sim. scores, as shown in the second row in Fig. 4 (b). (3) **MT (w/o OFT):** This experiment validates our progressive training philosophy by attempting to train all tasks (R2AV, RV2AV, RA2V) jointly from scratch without Omni-Task Fine-tuning (OFT). This naive multi-task (MT) approach yields suboptimal perfor-

mance on the R2AV task, particularly in text-following as indicated by ViCLIP (third row of Fig. 4 (b)). This confirms our hypothesis that when training a unified model, it is crucial to first establish a strong generative prior on weakly-constrained tasks (like R2AV) before introducing strongly-constrained tasks (like RV2AV/RA2V). Without this progression, the model tends to "shortcut" the learning process by overfitting to the easier, strongly-constrained tasks, ultimately failing to generalize on the more complex, weakly-constrained generation tasks.

# 5. Conclusion

In this paper, we present *DreamID-Omni*, a unified framework for controllable human-centric audio-video generation. By integrating reference-based generation, editing, and animation into a single paradigm, *DreamID-Omni* addresses the limitations of previous task-specific models. To tackle multi-person confusion, we introduce Syn-RoPE for signal-level identity-timbre binding and Structured Captioning for semantic disentanglement. Furthermore, Multi-Task Progressive Training harmonizes disparate objectives. Extensive experiments demonstrate that *DreamID-Omni* achieves state-of-the-art performance across multiple tasks.

## Acknowledgments

This work was supported in part by the National Natural Science Foundation of China under Project U23A20281.

## Impact Statement

This paper aims to advance controllable human-centric audio-video generation. However, it also raises risks related to unauthorized impersonation and deceptive media synthesis. Responsible use should involve consent from referenced individuals, transparent disclosure of generated content, and safeguards such as watermarking or misuse detection.

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

# A. Appendix

In the supplementary material, the sections are organized as follows:

- We provide qualitative comparisons with baselines on RV2AV and RA2V in Sec. A.1.

- We provide the details of our data construction pipeline in Sec. A.2.

- We provide the MLLM-based judge prompt in Sec. A.3.

- We provide more details regarding the user study in Sec. A.4.

- We discuss the limitations of our method and future work in Sec. A.5.

- We provide more qualitative results for R2AV, RV2AV, and RA2V tasks in Sec. A.6.

## A.1. Comparison Results on RV2AV and RA2V

Figs. 5 and 6 compare *DreamID-Omni* with state-of-the-art methods on RV2AV and RA2V, respectively, where the qualitative results clearly demonstrate our superior performance.

## A.2. Data Construction Details

Our full dataset consists of approximately 1M high-quality audio-video pairs. As illustrated in Fig. 7, our data construction pipeline is categorized into two primary stages: **In-pair data construction.** We process each video clip to extract its internal references. The reference voice timbre set $\mathcal{A}$ is created by applying DiariZen (Han et al., 2025) for speaker diarization to obtain precise timestamps. Concurrently, the reference identity set $\mathcal{I}$ is formed by using DWPose (Yang et al., 2023) to detect and crop face regions from keyframes. **Cross-pair data construction.** For the audio branch, we construct the reference timbre $\mathcal{A}$ through a multi-stage pipeline. First, DiariZen (Han et al., 2025) and Gemini (Team et al., 2023) are combined to accurately label speaker segments in multi-person dialogues. Subsequently, CosyVoice (Du et al., 2024) is employed to clone a clean voice for each speaker, which is then purified using ClearerVoice (Zhao et al., 2025b) for final denoising. For the video branch, the reference identity $\mathcal{I}$ is constructed following the established Phantom-Data (Chen et al., 2025d) pipeline.

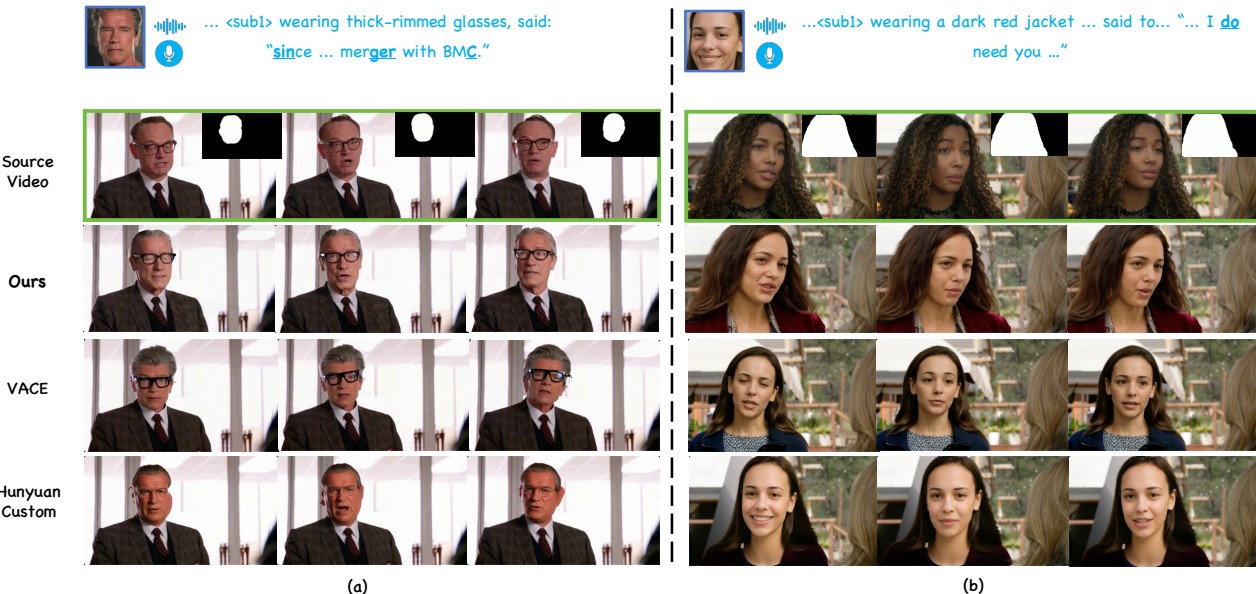

*Figure 5.* **Qualitative comparison** with state-of-the-art methods on RV2AV. Please zoom in for more details.

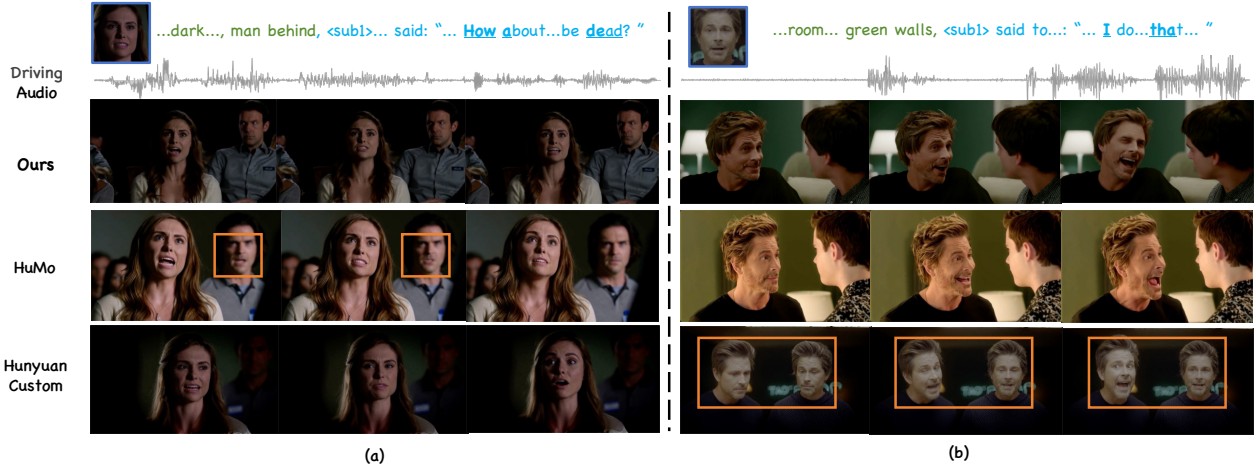

*Figure 6.* **Qualitative comparison** with state-of-the-art methods on RA2V. Please zoom in for more details.

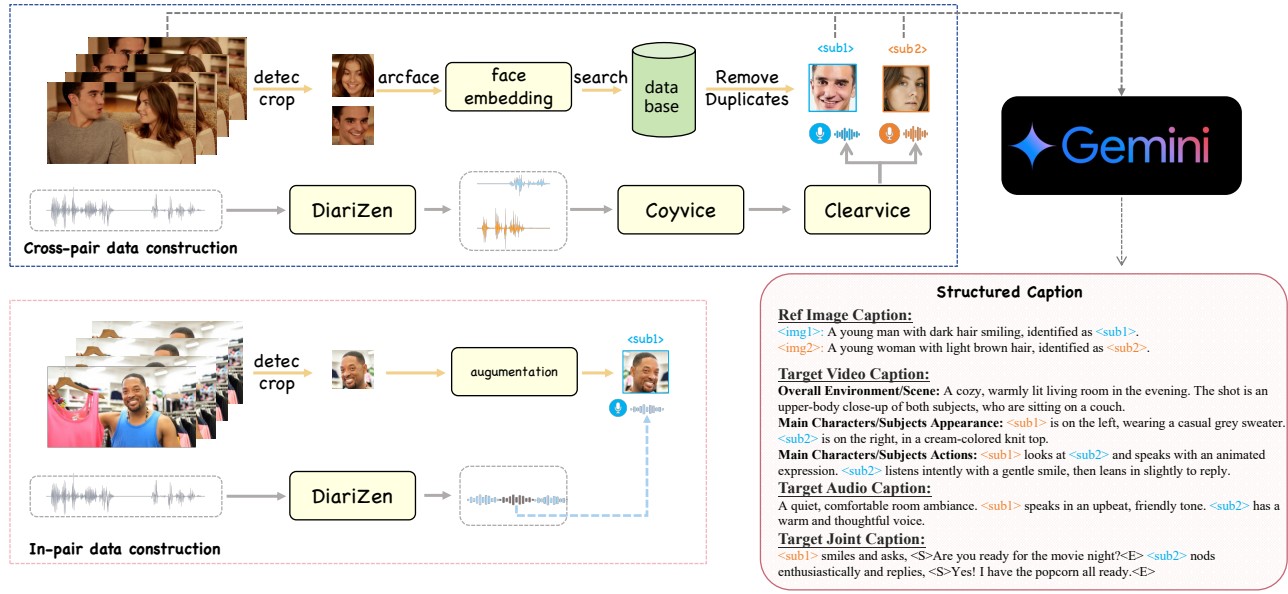

*Figure 7.* Data construction pipeline.

### A.3. MLLM-Based Judge

We employ Gemini-2.5-Pro as a MLLM-based judge for Speaker Confusion, Fig. 8 presents the system prompt.

### A.4. User Study

We conduct a user study as part of the evaluation on IDBench-Omni. Specifically, we invited 30 professional video creators to serve as evaluators. Users rate each video on seven dimensions on a 1–5 scale, and we average the ratings to obtain the final scores. The user study was carried out in a blinded setting. Table 7 indicates that our approach performs strongly across multiple dimensions.

### A.5. Limitations and Future Work

We discuss two main limitations of *DreamID-Omni* and outline corresponding directions for future work.

**Long-video generation.** Limited by the base model, *DreamID-Omni* currently supports videos up to 10 seconds, where the latent length stays safely below the Syn-RoPE margin $M = 150$. We leave minute-long generation to future work, where

**Role:**

You are a professional forensic video analyst specializing in multi-modal human-centric synthesis. Your task is to detect **Speaker Confusion** in generated videos where multiple identities and audio streams are present.

**Input Data:**

- Generated Video
- Reference Images
- Structured Caption

**Task Definition:**

Given a generated video, a set of reference images (ID images), and a structured caption, you must determine if the "Speaker Attribution" is correct. **Speaker Confusion (Output: 1)** occurs if the following conditions are met:

- Subject A is performing the lip movements or facial expressions corresponding to the dialogue lines assigned to Subject B in the caption.
- Subject A's visual appearance (e.g., clothing, face) is partially or fully blended with Subject B while they are speaking.

**Output Format:**

- **1** if Speaker Confusion is detected
- **0** if the video is consistent with the references and caption.

*Figure 8.* MLLM system prompt for speaker confusion detection.

*Table 7.* User Study with state-of-the-art methods on R2AV. Best results are in **bold**, second best are underlined.

| Method | Text-Video Alignment | ID-Sim. | Video Quality | Text-Audio Alignment | Timbre-Sim. | Audio Quality | Lip-sync |
|---|---|---|---|---|---|---|---|
| Phantom | 3.62 | 3.55 | 3.35 | - | - | - | - |
| VACE | 3.45 | 3.47 | 3.28 | - | - | - | - |
| Qwen-Image + LTX-2 | 3.32 | 3.09 | 3.14 | 4.18 | 2.41 | 3.73 | 2.91 |
| Qwen-Image + Ovi | 3.70 | 3.05 | 3.64 | 4.23 | 2.41 | 3.77 | 3.32 |
| Wan2.6 | 3.51 | 3.18 | **3.77** | 3.57 | 2.95 | 4.08 | 3.12 |
| **Ours** | **3.86** | **3.95** | 3.68 | **4.75** | **3.50** | **4.23** | **4.50** |

we plan to explore streaming architectures and dynamic RoPE designs.

**Inference efficiency.** To achieve strong controllability across heterogeneous conditions, our multi-condition Classifier-Free Guidance requires three forward passes per denoising step, which inevitably increases inference cost. To mitigate this overhead, we adopt sequence parallelism and CFG parallelism. As future work, we plan to explore DMD-style distillation to compress the multi-condition CFG into a single forward pass and further reduce the number of denoising steps.

## A.6. More Visual Results

As shown in Fig. 10-13, we provide more qualitative results of *DreamID-Omni* on R2AV, RV2AV and RA2V task.

**Role:**
You are specialized in the deep understanding and joint analysis of input image and user prompt. Your core responsibility is to act as a professional prompt engineering, merging complex visual and user prompt into a coherent, insightful caption for joint video-audio generation.

**Input Data:**
1. Reference Images: One or more images, typically containing one or more persons.
2. User Prompt: The text input by the user serves as a prompt for the joint generation of audio and video, including basic visual descriptions, and may also includes audio descriptions and speech content descriptions,The content of the speech is marked in double quotation marks.

**Task Definition:**
Your core objective is to generate a structured output containing the following four parts:
1. Ref Image Caption:
* Generate a concise description for each reference image Within 20 words in English. Focus on identifying and describing the main persons or subjects in the images
* Use a special tagging format to identify images and subjects, like `<imgN>` and `<subN>`, where `N` is a sequential number starting from 1.
2. Target Video Caption:
* Optimize based on the provided Reference Images and User Prompt.
* The primary task is to identify and replace all descriptive terms in the "user prompt" that refer to subjects matching those in the reference images with the corresponding special subject tag (e.g., `<sub1>`).
* Utilize your understanding of the reference images and user prompt to refine the visual description part of the "user prompt"**, ensuring its accuracy, richness of detail, and relevance to the reference images.
*. Please strictly follow this description order: Overall Environment/Scene, Main Character Appearance, Main Character Actions
3. Target Audio Caption:
* Optimize based on the provided Reference Images and User Prompt.
* Auditory Description (Non-Speech Section): If no Auditory Description is provided in the user prompt, then create a reasonable concise audio description and character emotions, voice descriptions based on the Target Video Caption.
4. Target Joint Caption:
* Optimize based on the provided Reference Images and User Prompt.
* Please strictly follow this description order: Main Character Actions and Speech content descriptions
* Speech content descriptions: please use the format `pure speech content<E>`. Don't delete or modify the the content of the speech in user prompt.

**Example output prompt:**
"Ref Image Caption: <img1>: A man with dark hair..., identified as <sub1>. <img2>: A smiling woman with dark hair..., identified as <sub2>. Target Video Caption: A coffee shop with warm ... ,<sub1> is on the left, wearing dark... <sub2> is on the right, wearing a simple...,<sub1> looks at ... <sub2> smiling... 、
Target Audio Caption: The gentle murmur of coffee shop chatter and the clinking of cups... Target Joint Caption: <sub1> ... says, I was hoping...<E> <sub2>... replies, Me too. It's ...<E>"
"""

*Figure 9.* MLLM system prompt for Structured Caption.

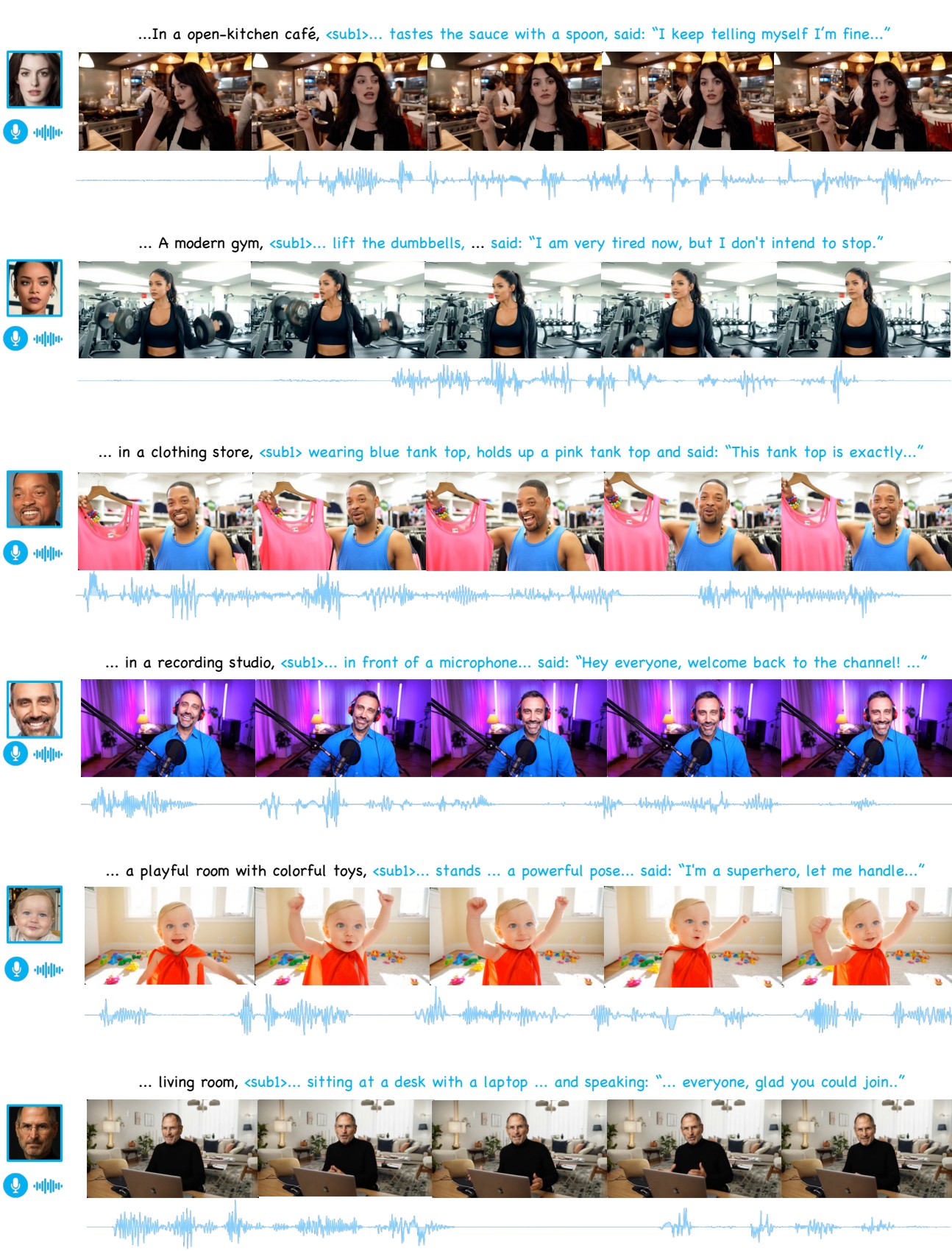

*Figure 10.* More **qualitative results on R2AV I**

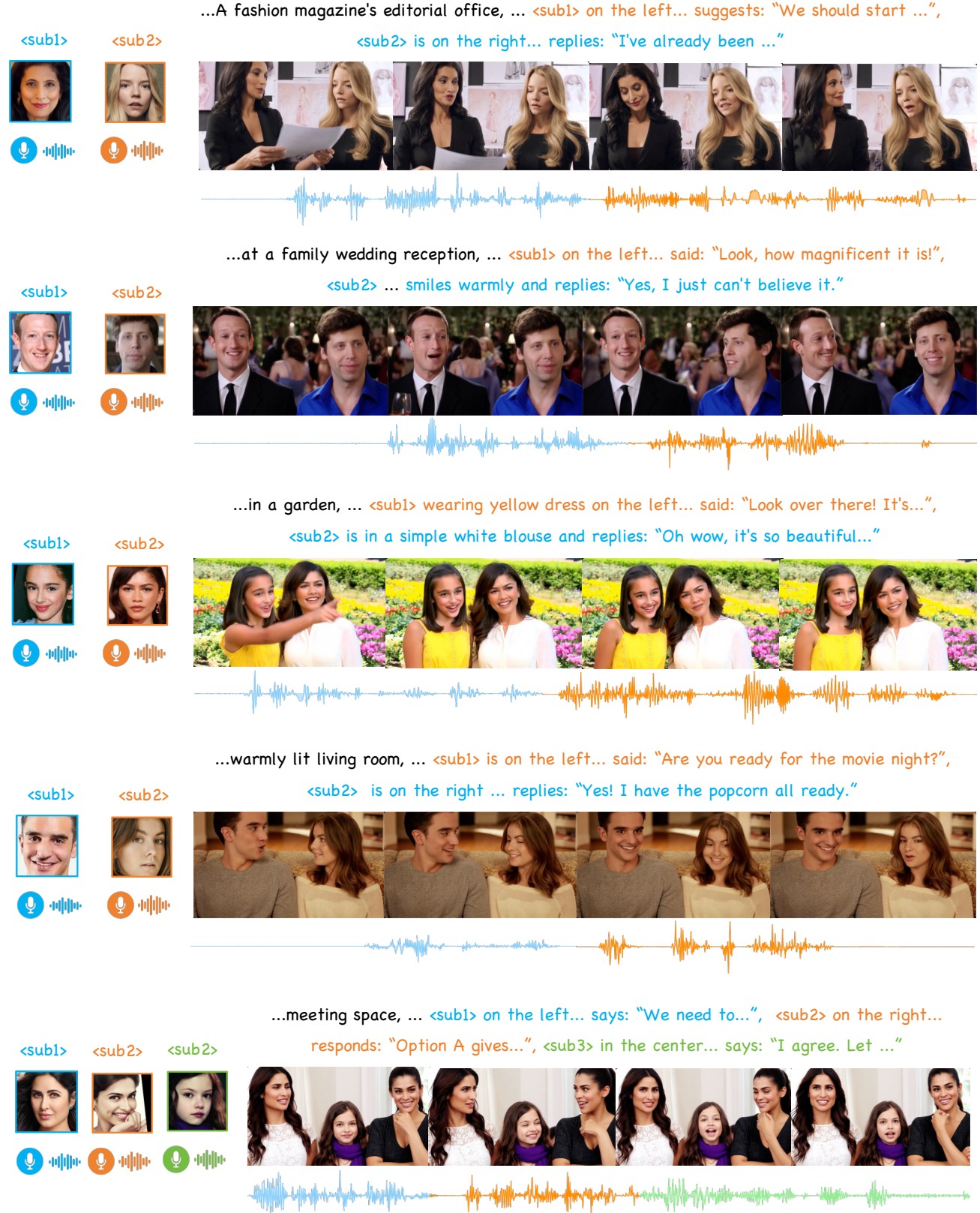

*Figure 11.* More **qualitative results on R2AV II**

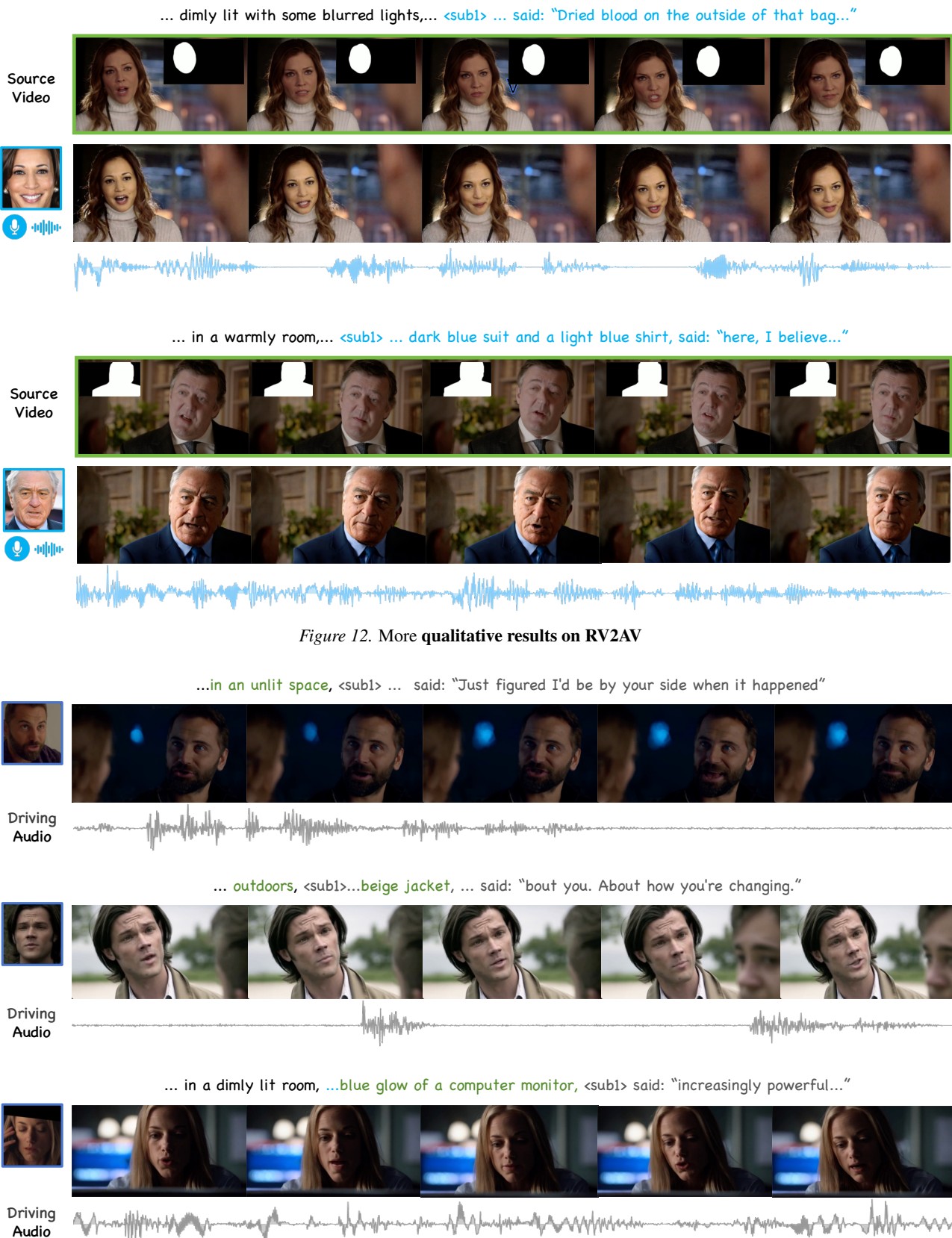

*Figure 12.* More **qualitative results on RV2AV**

*Figure 13.* More **qualitative results on RA2V**

