# OpenReview forum: "DreamID-Omni: Unified Framework for Controllable Human-Centric Audio-Video Generation"
_ICML.cc/2026/Conference — ICML 2026 regular_

### Official Review · Reviewer_rj16 · 2026-03-10

**Soundness:** 3
**Presentation:** 3
**Significance:** 3
**Originality:** 3
**Overall Recommendation:** 4
**Confidence:** 4

**Summary:**

The paper introduces DreamID-Omni, a unified framework for controllable human-centric audio-video generation. The authors observe that the field treats Reference-based Audio-Video Generation (R2AV), Video Editing (RV2AV), and Audio-Driven Animation (RA2V) as isolated tasks. To resolve this, they propose a Symmetric Conditional Diffusion Transformer (DiT) that handles static identity references (visual/acoustic) and dynamic structural constraints (source video/driving audio) within a single probabilistic mapping space. To mitigate speaker confusion in multi-subject scenes, the framework introduces a Dual-Level Disentanglement strategy: Synchronized Rotary Positional Embeddings (Syn-RoPE) at the continuous signal level, and Structured Captions parsed via an MLLM at the discrete semantic level. Optimization is stabilized via a Multi-Task Progressive Training curriculum.

**Compliance With Llm Reviewing Policy:**

Affirmed.

**Final Justification:**

I have carefully reviewed the author's rebuttal and the assessments provided by other reviewers. I am maintaining my original rating of Weak Accept. The authors clarified the use of Flow Matching and provided a convincing efficiency analysis. Regarding the divergence between reviewers, I believe the authors have sufficiently defended the importance of task unification by demonstrating 'cross-task' capabilities (e.g., lip-sync preservation during editing) that simpler baselines lack. While Reviewer 3EDB raises valid points on multi-person complexity, the author's new ablation on the margin $M$ addresses the primary technical risk for current generation lengths.

**Key Questions For Authors:**

- How does the model handle long-form video generation where the target sequence length $L$ exceeds the hard-coded Syn-RoPE margin $M$? Are there plans for dynamic RoPE scaling relative to sequence length?
- Can you provide a FLOPs or wall-clock efficiency analysis of the multi-condition CFG formulation, and what strategies could mitigate this 3x forward-pass overhead?
- How severely does the reliance on an external MLLM for Structured Captions impact total end-to-end latency?
- How well does the framework generalize to non-human or highly stylized subjects, given the reliance on ArcFace and DWPose for evaluation and potential data curation?
- Have you experimented with a truly unified tokenization scheme (where audio and video occupy the exact same dimensional latent space) to eliminate the $\gamma$ interpolation factor, or transitioning the objective from standard diffusion to Flow Matching?

**Limitations:**

No, the authors have not adequately discussed the computational limitations introduced by their triple-pass multi-condition guidance. They also fail to address the rigid limitations of the hard-coded Syn-RoPE margin $M$ when attempting to scale the framework to long-form cinematic generation.

**Strengths And Weaknesses:**

Strengths:
- The model effortlessly transitions between unconstrained generation and strict editing without requiring adapter layers or task-specific routing by concatenating reference features and injecting structural constraints via element-wise addition
- The three-stage curriculum (In-pair Reconstruction, Cross-pair Disentanglement, Omni-Task Fine-tuning) is a necessary approach that prevents the model from shortcutting to strongly constrained tasks and neglecting its core generative prior.
- The framework convincingly achieves state-of-the-art identity preservation and text-following against leading commercial and open-source models (Wan2.6, Ovi, HunyuanCustom)

Weaknesses:
- The Syn-RoPE mechanism relies entirely on a fixed, hard-coded temporal margin $M$. If a user attempts to scale generation to a long-form sequence where the latent length $L$ exceeds $M$, the rotational embeddings will collide with the reserved identity slots, causing catastrophic hallucination.
- Relying on an external, closed-source MLLM (Gemini-2.5-Pro) to preprocess unstructured prompts into Structured Captions introduces massive inference latency and API dependency. Furthermore, any hallucination by the MLLM propagates flawlessly into the DiT output.
- The multi-condition Classifier-Free Guidance (CFG) strategy requires three separate forward passes per denoising step. This essentially increases computational cost.
- The architecture inherits Ovi's asymmetric temporal resolutions, requiring a mathematically patched frequency scaling factor $\gamma = L_v / L_a$. Furthermore, it relies on standard $\epsilon$-prediction diffusion objectives rather than modern, highly efficient Flow Matching trajectories.

---

> ### Author Rebuttal · Authors · 2026-03-30
>
> We express our sincere gratitude to the reviewer for the insightful comments and the recognition of our work. We have addressed the specific concerns raised by the reviewer as detailed below.
>
> **W1, Q1, and L2: Long video and Syn-RoPE Scalability**
>
> Due to the current base model, our framework currently supports videos of up to 10 seconds. In this setting, the latent sequence length is 61, which remains well below the reserved margin for the first identity (M = 150). Therefore, the positional collision described by the reviewer does not occur within the duration regime studied in this work. **[Figure 2 at the link](https://github.com/icml2026sub2747-new/rebuttal/blob/main/rebuttal.md#figure-2-long-video-generation-examples-up-to-10s)** shows qualitative results on 10-second videos, and **[Table 2 at the link](https://github.com/icml2026sub2747-new/rebuttal/blob/main/rebuttal.md#table-2-ablation-m)** further demonstrates the ID stability of Syn-RoPE with M = 150.
>
> We agree that extending the framework to longer-form generation is an important future direction. In particular, we plan to explore streaming architectures and more flexible dynamic RoPE designs under autoregressive formulations.
>
> **W2 and Q3: Structured Caption Generation**
>
> During training, we require semantically reliable annotations for video understanding. We therefore use Gemini-2.5-Pro as an annotation tool, as it showed fewer hallucinations in our pilot study. Since speaker-content alignment is particularly error-prone, we further use DiariZen for speaker diarization during data construction to reduce hallucinations in speaker-content correspondence.
>
> At inference time, our framework does not rely on Gemini-2.5-Pro. The task is substantially simpler, as it only requires generating a structured caption from the reference images and the user prompt. This step can be handled by an open-source model, Qwen3-VL-8B-Instruct, with an average latency of about 14 seconds on a single A100 GPU.
>
> **W3, Q2, and L1: Inference Cost of Multi-Condition CFG**
>
> To achieve strong controllability and generation quality, we use three forward passes per denoising step in our multi-condition CFG formulation. We agree that this design increases inference cost.
>
> To mitigate this overhead, we implement several parallelization strategies, including sequence parallelism and CFG parallelism. Their runtime performance on A100 is summarized below:
>
> | Setting | Resolution | Time / iteration |
> |---|---|---|
> | multi-condition CFG | 512 x 992 | 7.41 s/it |
> | sequence parallelism = 2 | 512 x 992 | 4.71 s/it |
> | CFG parallelism = 2 | 512 x 992 | 4.11 s/it |
>
> Nevertheless, we agree that there is still room for further acceleration. As future work, we plan to explore DMD distillation to compress the multi-condition CFG into a single forward pass and further reduce the number of denoising steps.
>
> **W4 and Q5: Tokenization and Flow-Matching Formulation**
>
> Recent audio-video generation models, such as LTX-2 and Ovi, use separate VAEs to encode audio and video, followed by a scaling factor for alignment. We follow this design because it allows us to leverage existing high-quality VAEs while keeping training cost manageable. We agree that unified tokenization is a promising direction toward more native multimodal modeling, but it would likely incur higher training cost.
>
> We would also like to clarify that our model is trained as a velocity predictor under a flow-matching formulation, and UniPC is used as the solver during inference. We will make this point more explicit in the paper to avoid confusion.
>
> **Q4: Generalization Beyond Standard Human Subjects**
>
> Although our data construction pipeline relies on ArcFace and DWPose, the proposed framework is not intrinsically restricted to standard human subjects. We include qualitative results on non-human and highly stylized characters in **[Figure 5 at the link](https://github.com/icml2026sub2747-new/rebuttal/blob/main/rebuttal.md#figure-5-generalization-to-highly-stylized-subjects)**, which provide encouraging evidence of generalization beyond the primary human-centric setting.

---

> > ### Author Rebuttal · Reviewer_rj16 · 2026-04-02
> >
> > The rebuttal addresses my main concerns, so I will maintain my current score.

---

> > > ### Author Response · Authors · 2026-04-02
> > >
> > > Thank you for your time and your constructive comments. We appreciate your response and the positive evaluation of our paper.

---

### Official Review · Reviewer_Tvhm · 2026-03-11

**Soundness:** 3
**Presentation:** 3
**Significance:** 3
**Originality:** 3
**Overall Recommendation:** 5
**Confidence:** 4

**Summary:**

This paper proposes a unified DiT framework for controllable human-centric audio-video generation that handles three previously separate tasks—reference-based generation(R2AV), identity-aware video editing(RV2AV), and audio-driven animation(RA2AV) within one model. Its main technical ideas are a symmetric conditioning design for mixing identity, timbre, source-video, and driving-audio inputs; a dual-level disentanglement strategy combining synchronized RoPE for binding identities to voices and structured captions for reducing subject/attribute confusion in multi-person scenes; and a progressive multi-task training curriculum that first learns weakly constrained generation before adding stronger editing/animation objectives. The paper reports strong results on a new benchmark, IDBench-Omni, showing competitive or state-of-the-art performance across video quality, audio quality, lip-sync, and identity/timbre consistency, including in multi-person settings.

**Compliance With Llm Reviewing Policy:**

Affirmed.

**Key Questions For Authors:**

Please address my concerns in 'Weaknesses'.

**Limitations:**

yes

**Strengths And Weaknesses:**

Strengths

1. We've seen closed-source models like Wan2.6 and Seedance2 moving toward unified models for audio-video generation, reference-base generation and audio-driven generation. The research and open-source community has been a bit behind the curve here. I'm glad to see this work stepping up to explore this direction, especially by building on open-source foundations like OVI and the OpenHumanVid dataset.
2.  The core technical contributions—namely symmetric conditioning, dual-level disentanglement, multi-task progressive training, and the data construction pipelines—are well-motivated and technically sound.
3. The experiments are quite comprehensive and effectively back up the claims made about the proposed method's performance.

Weaknesses
1. In Table 2, the WER for Wan2.6 looks a bit suspicious. It’s unexpectedly poor, especially when you consider that its other audio and audio-visual metrics stay within a much more reasonable range. It would be helpful if the authors could double-check this or offer some context on why this specific metric is so low.
2. When comparing against other methods, were the prompts tailored or optimized for each specific baseline? My concern is that the highly structured prompts used in this work might not play well with other models. Clarifying whether the baselines were given prompts suited to their own training would help clear this up.

---

> ### Author Rebuttal · Authors · 2026-03-30
>
> We sincerely thank the reviewer for the careful reading and constructive comments. Below we address each concern in turn.
>
> **W1:**
> We double-checked the Wan2.6 results, and the reported WER is correct. A common failure case is that, although the caption specifies English speech, Wan2.6 often generates Chinese speech instead (**see Fig. 4 at the link(https://github.com/icml2026sub2747-new/rebuttal/blob/main/rebuttal.md#figure-4-wan26-failure-cases)** ). This explains why metrics such as CLAP and lip-sync-related scores remain in a reasonable range, while the language-dependent WER drops substantially. To improve reproducibility, we will release the benchmark together with the generated outputs of each baseline.
>
> **W2:**
> We adapted the prompts to match each baseline’s official or recommended input format to ensure a fair comparison. We did not directly apply our structured prompt format to the baselines. Specifically, Qwen-Image was prompted using the Diffusers implementation and its corresponding prompt format (**see Fig. 3 at the link(https://github.com/icml2026sub2747-new/rebuttal/blob/main/rebuttal.md#figure-3-qwen-image-prompt-used-for-baseline-first-frame-generation)**). LTX-2 used the official `gemma_i2v_system_prompt`. Ovi followed the official `example_prompts`, and Wan2.6 was prompted according to the format provided on its official website.

---

> > ### Author Rebuttal · Reviewer_Tvhm · 2026-04-02
> >
> > Thanks for the double check of Wan WER results and details of the prompt format.  I will keep my original score.

---

> > > ### Author Response · Authors · 2026-04-02
> > >
> > > We sincerely appreciate your positive feedback. Thank you for your time and valuable insights.

---

### Official Review · Reviewer_BJc6 · 2026-03-13

**Soundness:** 3
**Presentation:** 3
**Significance:** 3
**Originality:** 3
**Overall Recommendation:** 4
**Confidence:** 5

**Summary:**

The work focused on the task of Controllable Human-Centric Audio-Video Generation (conditioned on human reference images and speech), and proposed DreamID-Omni, a video generation model targeted at this task to achieve flexible video synthesis.
The paper also introduced a benchmark called IDBench-Omni (it would be better if it could be open-sourced) for the evaluation of this task.
The experiments display the SoTA performance of the proposed method.

**Compliance With Llm Reviewing Policy:**

Affirmed.

**Final Justification:**

The authors have addressed my concerns. I will maintain my positive score. There are some further suggestions:
- The RV2AV functionality is somewhat overclaimed if it is limited to the human face and body. Targeted revisions for the issue will be recommended.
- I also hope the benchmark could be fully released.

**Key Questions For Authors:**

Please see the weaknesses section.

**Limitations:**

yes

**Strengths And Weaknesses:**

### Strengths
- The presentation of figures is great in this paper.
- The paper is self-contained and easy-to-follow.
- The experiments are comprehensive and show the superior performance of the proposed method.
- Multi-Task Progressive Training is insightful, and it's also good to see the ablation study about it.
- The overall pipeline is solid and aligned with the industry standard of video generation applications.

### Weaknesses
- Cloning clean speech using CosyVoice and then further refining it may make the model better suited for clean and separable speech, but its robustness in the face of real-world speech is unknown.
- At training, the faces in the source video of the RV2AV task are always masked. I am afraid that it could lead to training-inference mismatching.
- It would be good if a visual geneartion paper could provide a human evaluation experiment.
- Some typos need to be fixed (e.g., "TasK" should be "Task" in Figure 2).

---

> ### Author Rebuttal · Authors · 2026-03-30
>
> We sincerely thank the reviewer for the positive assessment and constructive suggestions. Below we address each concern in turn.
>
> **W1:**
> We thank the reviewer for raising this important point. Although we use CosyVoice followed by ClearerVoice in the second-stage cross-pair pipeline to obtain cleaner reference timbre signals and reduce copy-paste artifacts, the model is not trained only on clean speech. In the first-stage in-pair reconstruction, the reference timbre is extracted directly from real videos without denoising. Moreover, the reference speech used in our benchmark (**see supplementary material Anonymous GitHub link**) is also taken from real-world recordings without denoising, and the corresponding quantitative results are reported in **Table 2** in the paper.
>
> **W2:**
> We thank the reviewer for this comment. For RV2AV, the source video is masked at inference in the same manner as during training. Therefore, under our RV2AV setting, there is no train-test mismatch in this respect.
>
> **W3:**
> We thank the reviewer for this suggestion. We have already included a human evaluation in Appendix A.4 (User Study), where 30 professional video creators served as evaluators in a blinded setting. We will highlight this more clearly in the main text.
> | Method | TV Alignment | ID-Sim. | Video Quality | TA Alignment | Timbre-Sim. | Audio Quality | Lip-sync |
> |---|---:|---:|---:|---:|---:|---:|---:|
> | Phantom | 3.62 | 3.55 | 3.35 | - | - | - | - |
> | VACE | 3.45 | 3.47 | 3.28 | - | - | - | - |
> | Qwen-Image + LTX-2 | 3.32 | 3.09 | 3.14 | 4.18 | 2.41 | 3.73 | 2.91 |
> | Qwen-Image + Ovi | _3.70_ | 3.05 | 3.64 | 4.23 | 2.41 | 3.77 | 3.32 |
> | Wan2.6 | 3.51 | 3.18 | **3.77** | 3.57 | _2.95_ | _4.08_ | 3.12 |
> | **Ours** | **3.86** | **3.95** | _3.68_ | **4.75** | **3.50** | **4.23** | **4.50** |
>
>
> **W4:**
> We thank the reviewer for pointing this out. We will correct this typo and carefully proofread the final version for similar issues.

---

> > ### Author Rebuttal · Reviewer_BJc6 · 2026-04-03
> >
> > Thanks for the responses. My concerns are mainly solved (good to see the human evaluation results). I have three more questions that need you to further clarify or explain:
> > 1. Could you explain the ratio of generated audio to real audio in the data? How do you ensure the high quality of the real audio?
> > 2. So, in the RA2AV inference, the proposed model also requires masking the human face for face swapping? More explanation is welcome if I am wrong. Also, is the RA2AV task limited to face swapping?
> > 3. Just saw some visual artifacts in the uploaded videos of your supplementary material. Does it have any additional techniques or mechanisms to reduce these (apart from data-related methods)?

---

> > > ### Author Response · Authors · 2026-04-04
> > >
> > > We sincerely thank the reviewer for the follow-up response and constructive suggestions.
> > >
> > > **Q1:**
> > >
> > > - The ratio of real data to synthetic data is approximately 3:1. With more real data (in-pair), the model tends to copy the background noise from the reference audio. while with more synthetic data (cross-pair), the timbre quality degrades. This ratio provides a good trade-off.
> > >
> > > - To ensure the quality of real audio, we mainly focus on two criteria when filtering the data sources: audio signal-to-noise ratio (SNR) and audio-visual synchronization. Specifically, SNR (> 50 dB) is used to assess audio quality, while Sync-C (> 3) is used to remove samples with excessive background noise or poor synchronization.
> > >
> > > **Q2:**
> > >
> > > - The reviewer’s understanding is correct. In RV2AV, a mask is provided as input. The editable region is not limited to the face, but can also correspond to the head or the full human region in the frame. Such a mask-based formulation avoids the need to construct additional explicit paired supervision, which in turn enables the three tasks to be unified using the same source data. Qualitative RV2AV results are provided in Figure 5 and Figure 13 of the paper or at the following link (https://github.com/icml2026sub2747-new/rebuttal/blob/main/rebuttal.md#figure-6-rv2av-visualization).
> > >
> > > **Q3:**
> > >
> > > We thank the reviewer for the careful reading. Artifacts remain a challenging issue across the entire field of generative modeling. Beyond data filtering, we also make efforts at both the training and inference stages to reduce artifacts.
> > >
> > > - At the training stage, an important source of artifacts is the interference among different conditions across the three tasks, including the reference information, source video, and driving audio. Our Symmetric Conditioning balances heterogeneous signals, while the Multi-task Progressive Training strategy effectively alleviates such interference, thereby reducing artifacts and improving generation quality (Table 6 and Figure 4 in the paper).
> > >
> > > - At the inference stage, following Humo[1], dynamically adjusting the guidance scales of multi-condition CFG over the denoising steps helps improve generation quality. Specifically, we increase the guidance from text conditions in the early denoising stage, and place stronger guidance on the reference information in the later stage.
> > >
> > > However, we agree that artifacts are still not fully resolved. In future work, we plan to further alleviate this issue by constructing preference pairs with and without artifacts, and exploring DPO- or RL-based optimization methods.
> > >
> > > We thank the reviewer for the positive assessment of our work. We hope that the above responses have addressed the reviewer’s concerns.
> > >
> > > [1] HuMo: Human-Centric Video Generation via Collaborative Multi-Modal Conditioning. AAAI 2026

---

### Official Review · Reviewer_3EDB · 2026-03-14

**Soundness:** 2
**Presentation:** 3
**Significance:** 1
**Originality:** 3
**Overall Recommendation:** 2
**Confidence:** 5

**Summary:**

This paper proposes DreamID-Omni, a unified framework for controllable human-centric audio-video generation that jointly handles three tasks: reference-based audio-video generation (R2AV), reference-based video editing with audio generation (RV2AV), and reference-based audio-driven video animation (RA2V). The method is built on a dual-stream DiT with a Dual-Level Disentanglement design consisting of Syn-RoPE for signal-level identity/timbre binding and Structured Captions for semantic grounding. The paper only evaluates quantitatively the method on their newly introduced IDBench-Omni benchmark.

**Compliance With Llm Reviewing Policy:**

Affirmed.

**Final Justification:**

After rebuttal, Answers to W1, W2, L2 still far to address concerns. Modify the motivation and claims carefully with more reproduction details would make a more strong and solid paper.

**Key Questions For Authors:**

see weakness.

**Limitations:**

No. Some limitations:
(1) long-term generation. Ovi already supports 10s, but most shown results are up to only 5s, especially for RA2V(namely talking avatar task is quite short)
(2) Lack details for Qwen-Image + LTX-2/Ovi baselines which some concerns on the implementation are raised. we can see in Fig 3. (b), If you use Qwen-Image to generate the first frame, why the Jobs not appear at all? Which type of prompt did you use for these t2av methods, same as for your method?

**Strengths And Weaknesses:**

Strength:
1. Best results on their benchmark and user study.
2. Reasonable techniques for solving multi-speaker confusion.

Weakness:
1. The paper mixes two different notions of unification, but only one seems truly important.
In my view, the more meaningful problem is multi-person identity/timbre disentanglement, rather than unifying R2AV, RV2AV, and RA2V into a single framework. The task-level unification itself does not appear sufficiently important, and the proposed technical route is not particularly novel.
The paper does not compare against simpler alternatives. For example, it is plausible that training only on R2AV, together with region-specific diffusion forcing strategy then partial denoising at inference time (or even R2AV training only then partial denoising at inference), could yield reasonable RV2AV behavior. Without such comparisons, the claimed benefit of explicitly unifying the three tasks is not well established. In addition, for RV2AV, since all region is noised and generated, the paper should evaluate how much non-face regions are unnecessarily changed, which is an important criterion for an editing task.

2. The more meaningful multi-person setting is still insufficiently analyzed.
The structured subject tags (e.g., <sub1>) may not be a very natural interface for T5 embeddings, yet this design choice is not carefully examined. For Syn-RoPE, the paper does not ablate the margin parameter M, nor does it analyze how identity and timbre consistency change as the number of people increases. This is important because increasing the number of subjects also increases rope index dramatically, which could weaken attention and make later identities harder to preserve. Without such analysis, the core claim on robust multi-person binding remains incomplete.

3. The evaluation relies too heavily on an in-house benchmark.  For RA2V(namely talking avatar), it lacks many true SOTA methods and the result on a public benchmark. For Timbre, should provide comparison to TTS methods for reference.

4. Lack enough details for reproduction, such as final dataset scale for training, training resource and time, lora or full-fintuing, infercence setting and so on.

---

> ### Author Rebuttal · Authors · 2026-03-30
>
> We thank the reviewer for the constructive comments and suggestions.
>
> **W1:**
> - We agree that multi-person identity/timbre disentanglement is a central technical challenge, but we would like to further clarify the significance of unifying R2AV, RV2AV, and RA2V.
>   - Unified training enables **cross-task capabilities**. For example, in the second RV2AV example ("messi") in **Fig. 1 in paper**, our model can follow driving audio while editing the video, so that the edited result remains lip-synchronized with the original performance. This addresses an important challenge in video editing: preserving lip-sync consistency after visual edits, which traditionally requires explicit supervision but is achieved in our unified setting using only masked videos.
>   - We also believe task unification has clear **practical importance**. Unifying multiple tasks into a single framework is valuable because it reduces the cost of acceleration and deployment in real applications. Our **novelty** lies in recognizing the structural symmetry among these three tasks: they rely on the same source data, but differ in conditioning strength. By exploiting this shared structure, we achieve unified modeling with **very low additional data and training cost**.
>   - We tested the simpler alternative of "R2AV-only training then partial denoising at inference". As shown in **Fig. 1 at the link below**, it fails to produce satisfactory RV2AV results, which further supports the necessity of unified training.
> - We also evaluated preservation of non-edited regions. As shown in **Table 1 at the link**, DreamID-Omni better preserves non-edited regions.
>
> **W2:**
> - For the structured anchor tokens, our design needs to satisfy two conditions: (1) reducing ambiguity [1] when multiple references share similar high-level attributes (**see Fig. 4(a) in the paper**), and (2) keeping the token usage within the T5 budget `512`. Since the anchor only needs to be defined once at the beginning, it saves tokens in subsequent descriptions. In principle, any anchor token that satisfies these conditions should work.
> - We add an ablation on M (**[Table 2 at the link](https://github.com/icml2026sub2747-new/rebuttal/blob/main/rebuttal.md#table-2-ablation-m)**). The results show that, in our setting, `M = 150` provides a good tradeoff between identity consistency and identity-timbre binding. Owing to RoPE's extrapolation ability, the performance is not sensitive to M within a certain range. Due to the limitation of academic datasets, we can only obtain data with up to 3 subjects. To approximate settings with more subjects, we additionally scale `M` to `N x M` on single-subject data to mimic the `N`-subject case (**Table 3 at the link**), which provides supporting evidence that 4- or 5-subject settings do not weaken attention in our setting.
>
> **W3:**
> - To the best of our knowledge, DreamID-Omni is the first academic open-source model that can handle R2AV and RV2AV. Therefore, there is currently no public benchmark that directly covers this setting. To fill this gap, we introduced IDBench-Omni and will release it publicly.
> - For RA2V, we believe there may be **some ambiguity in task definition**. Our setting is not standard talking-avatar generation, which typically takes a complete first frame as input; instead, our model takes a reference ID, with the first frame generated by itself. To the best of our knowledge, only Humo and HunyuanCustom support a comparable RA2V setting. To further validate this task, we additionally report results on the public MoCha[2] benchmark and add comparisons with talking-avatar models (**Table 4 at the link**).
> - We also add comparisons with TTS methods under the R2AV setting (**Table 5 at the link**).
>
> **W4:**
> - Thank you for pointing this out. Our model is fully fine-tuned on 1M training samples using 32 H20 GPUs for 50k training steps, totaling about 128 GPU days. We will add these missing reproduction details to the paper.
>
> **L1:**
> - Our model supports video generation of up to **10s** for R2AV, RV2AV, and RA2V. Our benchmark also includes source videos and driving audio longer than 8s for evaluation, and we provide representative examples in **Fig. 2 at the link**.
>
> **L2:**
> - We follow the official Qwen-Image prompting recipe to generate the first frame (code in **Fig. 3 at the link**). In challenging cases where multiple identities are visually similar and the scene description is complex, Qwen-Image may confuse the subject. This also highlights the advantage of our end-to-end modeling over a multi-stage cascade pipeline. For Ovi and LTX-2, we also adapt the prompts according to their official input formats to ensure a fair comparison.
>
> [1] Movie Weaver: Tuning-Free Multi-Concept Video Personalization with Anchored Prompts. CVPR 2025
>
> [2] MoCha: Towards Movie-Grade Talking Character Synthesis. NeurIPS 2025
>
> All figures and tables are available at the **link**: https://github.com/icml2026sub2747-new/rebuttal/blob/main/rebuttal.md

---

> > ### Author Rebuttal · Reviewer_3EDB · 2026-04-04
> >
> > Thanks for the response, but W1, W2, L2 still not address my concerns.

---

> > > ### Author Response · Authors · 2026-04-04
> > >
> > > Thank you for your response. We would be very grateful if you could indicate more specifically which points in W1, W2, and L2 you believe remain unresolved. Such clarification would help us better understand your position.
> > >
> > > # **W1**:
> > >
> > > - The significance of unification lies in its **cross-task capabilities** (e.g., the second RV2AV example, "messi", in Figure 1 of the paper), its **practical importance**, and the fact that the simpler alternative mentioned by the reviewer does not work well ([Figure 1 at the link](https://github.com/icml2026sub2747-new/rebuttal/blob/main/rebuttal.md#for-compliance-with-the-icml-rebuttal-policy-videos-are-converted-into-figures)).
> > >
> > > - We evaluated preservation of non-edited regions ([Table 1 at the link](https://github.com/icml2026sub2747-new/rebuttal/blob/main/rebuttal.md#table-1-preservation-of-non-edited-regions-in-rv2av) or below), where DreamID-Omni better preserves non-edited regions.
> > >
> > > Table 1. Preservation of non-edited regions in RV2AV.
> > >
> > > | Method | Non-Edit Change@10 ↓ | Non-Edit MAE ↓ |
> > > | --- | ---: | ---: |
> > > | VACE | 0.094 | 0.032 |
> > > | HunyuanCustom | 0.113 | 0.037 |
> > > | Ours | **0.076** | **0.028** |
> > >
> > >
> > > # **W2:**
> > >
> > > - Structured Captions have been ablated in **Table 5 of the paper**. Compared with the captions used in our base model Ovi, they bring consistent improvements on both audio and video metrics.
> > >
> > > - We added ablation results on the margin parameter M ([Table 2 and Table 3 at the link](https://github.com/icml2026sub2747-new/rebuttal/blob/main/rebuttal.md#table-2-ablation-m) or below), which show that under our current setting, identity and timbre consistency do not deteriorate as the number of identities increases, and that more identities do not weaken attention or make later identities harder to preserve.
> > >
> > > Table 2. Ablation `M`
> > >
> > > | M | ID1-Sim ↑ | ID2-Sim ↑ | ID3-Sim ↑ | T-Sim ↑ |
> > > | --- | ---: | ---: | ---: | ---: |
> > > | Vanilla concat | 0.593 | 0.587 | 0.601 | 0.211 |
> > > | 100 | 0.603 | 0.605 | 0.597 | 0.384 |
> > > | 150 | 0.607 | 0.601 | 0.603 | 0.402 |
> > > | 300 | 0.598 | 0.611 | 0.604 | 0.397 |
> > > | 500 | 0.594 | 0.604 | 0.598 | 0.404 |
> > > | 1000 | 0.596 | 0.574 | 0.463 | 0.335 |
> > >
> > > Table 3. Evaluation for larger-subject settings by scaling margin `M`. absolute similarity is higher than the ~0.61 multi-subject level because the multi-subject benchmark contains more profile-view dialogue scenes.
> > >
> > > | Margin Setting | M | 3M |4M | 5M |
> > > | --- | ---: | ---: | ---: | ---: |
> > > | ID-Sim (Single-Subject) ↑ | 0.674 | 0.663 | 0.671 | 0.667 |
> > >
> > >
> > > # **L2:**
> > >
> > > - All baseline prompts were adapted to their official input formats to ensure a **fair comparison** (the Qwen-Image prompting code is shown in [Fig. 3 at the link](https://github.com/icml2026sub2747-new/rebuttal/blob/main/rebuttal.md#figure-3-qwen-image-prompt-used-for-baseline-first-frame-generation)). In addition, [results at the link](https://github.com/icml2026sub2747-new/rebuttal/blob/main/rebuttal.md#figure-7-baseline-results-of-r2iovi-and-r2iltx-2-in-identity-as-presence) reported by a recent paper[1] (published after our submission, but using a similar base model and similar baselines) also support the reliability of our evaluation.
> > >
> > > We would sincerely appreciate any more specific comments, which would help us better address your concerns.
> > >
> > > [1] Identity as Presence: Towards Appearance and Voice Personalized Joint Audio-Video Generation. Arxiv 2026.

---

### Decision · Program_Chairs · 2026-04-30

**Decision:**

Accept (regular)

**Comment:**

The paper proposes DreamID-Omni, a unified framework for controllable human-centric audio-video generation that jointly handles reference-based generation, video editing, and audio-driven animation. Key design choices include conditioning injection for multiple input signals, a mechanism to keep each person's identity and voice separate in multi-person scenes, and a staged training curriculum.

Scores - 2 (R), 4 (WA), 4 (WA), 5 (A)


Reviewers acknowledge several strengths including:
- Unified framework for three human-centric AV tasks (BJc6, Tvhm, rj16)
- State-of-the-art results on benchmark and user study (all reviewers)
- Training curriculum design (BJc6, rj16)
- Comprehensive experiments (BJc6, Tvhm)

The weaknesses noted by reviewers include:
- Scalability to longer videos and robustness of Syn-RoPE margin M (rj16, 3EDB)
- Baseline comparison fairness (Tvhm, 3EDB)
- Inadequate validation for Multi-person setting (3EDB)
- Limited technical novelty (3EDB)
- Insufficient reproducibility details (3EDB)

In the rebuttal, the authors provided additional results on a public benchmark, ablations on key design choices, evaluation of editing fidelity, reproducibility details, and an inference cost analysis. Three of four reviewers found the responses satisfactory, while 3EDB maintained that the multi-person evaluation evidence remains insufficient, and that the paper reads more as a strong system effort than a research contribution.

While 3 of 4 reviewers gave positive recommendations, 3EDB's concerns remained after engagement. However, rj16 and BJc6 both read 3EDB's critique and weighed it against the rebuttal evidence. rj16 found the new ablations to address the primary technical risk for current generation lengths, and BJc6 maintained a positive recommendation after considering the critique.

The Area Chair agrees with the majority of reviewers and recommends acceptance. In the camera-ready version, authors are advised to revise based on the discussions in the rebuttal, and to strengthen reproducibility details.